# Childhood Trauma and Psychological Distress: A Serial Mediation Model among Chinese Adolescents

**DOI:** 10.3390/ijerph18136808

**Published:** 2021-06-24

**Authors:** Lin Zhang, Xueyao Ma, Xianglian Yu, Meizhu Ye, Na Li, Shan Lu, Jiayi Wang

**Affiliations:** 1Key Laboratory of Adolescent Cyberpsychology and Behavior, Ministry of Education, Wuhan 430056, China; lin_zhang@ccnu.edu.cn (L.Z.); lushan@mails.ccnu.edu.cn (S.L.); 2Key Laboratory of Human Development and Mental Health of Hubei Province, School of Psychology, Central China Normal University, Wuhan 430056, China; 3School of Psychology, Central China Normal University, Wuhan 430056, China; 4Department of Psychosomatic Medicine and Psychotherapy, University of Ulm, Medical Psychology, Frauensteige 6, 89075 Ulm, Germany; xueyao_ma@hotmail.com; 5Department of Education, Jianghan University, Wuhan 430056, China; 15872393952@stu.jhun.edu.cn; 6Putian Huaqiao High School, Putian 351115, China; ptqzymz@hotmail.com; 7Lie Dong Middle School, Sanming 365000, China; fjsmldzx@gmail.com

**Keywords:** childhood trauma, psychological distress, social support, family functioning, serial mediation, adolescent

## Abstract

The consequence of childhood trauma may last for a long time. The purpose of the present study was to examine the effect of childhood trauma on general distress among Chinese adolescents and explore the potential mediating roles of social support and family functioning in the childhood trauma-general distress linkage. A total of 2139 valid questionnaires were collected from two high schools in southeast China. Participants were asked to complete the questionnaires measuring childhood trauma, social support, family functioning, and general distress. Pathway analysis was conducted by using SPSS AMOS 24.0 and PROCESS Macro for SPSS 3.5. Results showed that childhood trauma was positively associated with general distress among Chinese adolescents. Social support and family functioning independently and serially mediated the linkage of childhood trauma and general distress. These findings confirmed and complemented the ecological system theory of human development and the multisystem developmental framework for resilience. Furthermore, these findings indicated that the mental and emotional problems of adolescents who had childhood trauma were not merely issues of adolescents themselves, but concerns of the whole system and environment.

## 1. Introduction

Improving the mental health of adolescents is an important public health problem globally. According to World Health Organization (WHO) report, the prevalence of mental disorders among children and adolescents was 12–29% [1]. In China, the prevalence of behavioral and emotional problems of adolescents aged 12–16 was 23.1%, according to a recently published national survey [2]. The consequence of mental disorders, especially the emotional problems of adolescents, may be significant and may not only influence the individuals’ mental health in adulthood but also result in great challenges and burdens on their families and societies [1].

Even though every adolescent is at risk of developing mental disorders and emotional problems, some factors make an adolescent more susceptible to mental disorders or emotional problems. Childhood trauma is one of these factors. Childhood trauma is defined as a perceived experience in childhood that gives impends of injury, death, or physical integrity, and causes negative feelings (e.g., fear, terror, and helplessness) [3]. A large number of empirical studies have demonstrated the increased rates of mental disorders and emotional problems for individuals who experienced childhood trauma [4,5,6]. Nearly 30% of mental disorders or emotional problems are due to childhood trauma [7]. This effect of childhood trauma on mental health may be long-lasting, from the moment the trauma occurs to adulthood and old age [8].

However, it is noteworthy that not all adolescents who experience childhood trauma suffer from mental disorders and emotional problems. Many people who experience trauma in childhood do not show any symptoms or do not have long-lasting mental distress, in which resilience may play an important protective factor [9]. A recent review indicated that social support and family functioning might be important resilience factors of childhood adversity, however, the relationship between these resilience factors has not been fully examined [10].

Moreover, it is important for adolescents to investigate resilience factors and the interrelationships of these factors in the linkage of childhood trauma and mental health for the following reasons. First, adolescents are transferring their focus gradually from their family to peers, school, and community. Meanwhile, the range of social interaction is broadened during the period of adolescence compared with childhood, which indicates that adolescents may have chances to look for new resources and support to deal with their trauma and mental distress [11]. Moreover, adolescence is another critical period of rapid development of brain growth and function compared with adulthood and old age. Childhood trauma may change neural plasticity and function, while interventions during adolescence have the potential to reduce this negative consequence [8].

Therefore, the present study examined the effect of childhood trauma on adolescents’ mental health (i.e., general distress), explored some resilience factors with this association, and investigated the relationship between these resilience factors.

### 1.1. Theoretical Bases

The present study is based on the ecological system theory of human development and the multisystem developmental framework for resilience [12,13].

The ecological system theory of human development was proposed by Bronfenbrenner [12]. He believed that the development of adolescents was affected by a set of nested contexts (i.e., microsystem, mesosystem, exosystem, macrosystem), which indicated that influence factors of mental health of adolescents presented in the family, school, and community could be both part of the past and the present environments [12].

The multisystem developmental framework is a contemporary theory for resilience, in which theorists emphasize that resilience is not a stable permanent characteristic of individuals but a state of functioning at a particular time that reflects a grouping of individual characteristics, external supports, and current stressors [13,14].

### 1.2. Childhood Trauma and General Distress

A great number of previous studies have proven the effect of childhood trauma on general distress among adolescents. A cross-sectional study of Chinese adolescents found a positive association between childhood trauma and depressive symptoms [15]. Another previous study also found that juvenile offenders exposed to adverse childhood experiences were at risk for general distress [4].

The above findings can be explained by the following theories. As mentioned in the “kindling” hypothesis and sensitizing effect, some researchers consider that childhood trauma can heighten the sensitivity or increase the vulnerability of an individual to negative events, that is, minor events may induce a strong emotional response in the future [16,17].

Although many theories have proven the effect of childhood trauma on general distress, some researchers proposed an opposite view. For example, Rutter proposed the stress inoculation theory and the “steeling” effect, in which he considered that childhood or adolescence adversity sometimes can strengthen resistance to later stress. In other words, adversity in some circumstances can harden and improve an individual’s psychological and physiological ability, such as self-esteem, self-efficacy, and exercise habits [18,19]. 

The disagreements on the effect of childhood adversity on general distress indicated that some mediating variables might exist in the linkage of childhood adversity and general distress. Examining the association between childhood adversity and general distress and exploring the pathway in this relationship would help researchers get a better understanding of the potential mechanism. Besides, because most previous studies found a positive association between childhood trauma and general distress, it is reasonable to assume that this relationship would always appear unless adolescents are in a particular circumstance, or receive appropriate intervention or guidance. Because the participants in the present study did not receive any intervention or guidance, we hypothesized that childhood trauma would be positively associated with general distress among them.

### 1.3. The Mediating Role of Social Support

Childhood trauma can affect adolescents’ social support. According to the social support deterioration model, if individuals were exposed to a traumatic experience during their childhood, they may find it hard to trust others and may perceive less support in the future, further affecting the support they obtain and their ability to use support [20,21,22]. Besides, some previous studies also supported the association between childhood trauma and social support. These studies investigated the potential association among veterans and undergraduate students and found that childhood trauma was negatively associated with social support [20,23,24].

Social support has a negative association with general distress, which could be confirmed by a large number of previous studies [25,26]. For example, a literature review summarized the evidence of the negative association between social support and depressive symptoms [27]. A possible explanation for the association is that social support could be considered as interpersonal and emotional regulation. Negative emotions could be modified by interpersonal and emotional regulation via attentional deployment or cognitive change [27]. Therefore, we hypothesized that social support (e.g., numbers of friends who can get help) may mediate the association between childhood trauma and general distress.

### 1.4. The Mediating Role of Family Functioning

Childhood trauma may impair family functioning. According to the suboptimal environmental hazards model, people who experienced trauma in childhood are more likely to view caregivers as the source of terror and find it difficult to trust their family members, which negatively impacts the child–parent relationship and family functioning [28]. Besides the theoretical model, some cross-sectional surveys also found a negative association between childhood trauma and family functioning. On the contrary, a supportive family environment is an important resilience factor of childhood trauma. Previous studies found that adolescents who had experienced childhood trauma had higher rates of resilience if they received enough care and love in adolescence from their parents [29]. 

The effect of family functioning on general distress can be supported by both a theoretical model and empirical studies. According to the proximal environmental mitigation model, the family relationship is an adolescent’s proximal resource, which may be strongly associated with the mental health of adolescents [30]. A systematic review and meta-analysis found that family functioning was positively correlated with the psychological health of children and adolescents [31]. Based on the above theoretical and empirical findings, we hypothesized that family functioning (e.g., possibility of expressing feelings in the family) may mediate the linkage of childhood trauma and general distress among Chinese adolescents.

### 1.5. The Serial Mediating Role of Social Support and Family Functioning

Social support may change the family dynamic and functioning. According to the family life cycle theory, an ordinary Chinese adolescent’s family is relatively stable if there are not unpredictable and external stressors and challenges in the family [32]. However, if an individual is exposed to a traumatic environment (often in the family) in childhood, this individual may be capable of changing or disengaging from the traumatic environment during adolescence through other supports and resources (e.g., communities, teachers, psychological counselors, and peers) because the person has a wider social circle during adolescence. A previous study also found that the social support of family members could affect family functioning [33]. Therefore, we hypothesized that social support may be an antecedent of family functioning in the present study. Above all, we hypothesized that social support and family functioning may serially mediate the linkage of childhood trauma and general distress.

### 1.6. The Current Study

Overall, the purpose of the present study was to examine the effect of childhood trauma on general distress among Chinese adolescents, further exploring the pathway in the linkage of childhood trauma and general distress (i.e., the mediating roles of social support and family functioning in the association between childhood trauma and general distress). Our hypotheses are as follows:

**Hypothesis** **1** **(H1).**Childhood trauma is positively associated with general distress among Chinese adolescents.

**Hypothesis** **2** **(H2).**Social support plays a mediating role in the association between childhood trauma and general distress.

**Hypothesis** **3** **(H3).**Family functioning plays a mediating role in the association between childhood trauma and general distress.

**Hypothesis** **4** **(H4).**Social support and family functioning plays a serial-mediation role in the association between childhood trauma and general distress.

## 2. Materials and Methods

### 2.1. Participants

Participants were recruited from two high schools in southeast China by convenient sampling. The recruited participants had the option of completing an online or paper questionnaire. A total of 2300 questionnaires were distributed from 9–30 April 2021. Among them, 2139 (93%) questionnaires were valid. The response rate was 100% and the valid rate was 93%. The exclusion criteria were as follows: (1) questionnaires completed in less than 1 min or more than 20 min, (2) questionnaires completed less than 90%.

The present study was approved by the Life Science Ethics Committee of Central China Normal University. Participants were free to quit the study at any time without consequence. All the data were used only for research and collected on a strictly voluntary and anonymous basis. All participants gave their consent to participate in the present study.

### 2.2. Measures

#### 2.2.1. Outcome

General distress was considered as the dependent variable and measured using the Chinese version of Depression Anxiety Stress Scale 21 (DASS-21-C) [34]. The DASS-21-C is a 21-item, 4-point Likert scale (0 = *did not apply to me at all*, 3 = *applied to me very much, or most of the time*), which was adapted from the Depression Anxiety Stress Scale 21 [35]. The DASS-21-C measures general distress experienced over the prior week, which includes stress (e.g., “I found it hard to wind down”), anxiety (e.g., “I was aware of dryness of my mouth”), and depression (e.g., “I couldn’t seem to experience any positive feeling at all”). A high total score means a high level of general distress. In the present study, the internal consistency for the scale was good (Cronbach’s α = 0.96).

#### 2.2.2. Independent Variable

Childhood adversity was regarded as the independent variable and measured using the Chinese version of the Childhood Trauma Questionnaire (CTQ-SF-C) [36]. The CTQ-SF-C is a 28-item, 5-point Likert scale (1 = *never true*, 5 = *very often true*), which is adapted from the Childhood Trauma Questionnaire and measures traumatic experiences before 18 years of age [37]. The CTQ-SF-C includes five subscales, emotional abuse (e.g., “I felt hated by family”), physical abuse (e.g., “I was hit hard enough to leave bruises”), sexual abuse (e.g., “I was touched sexually”), emotional neglect (e.g., “family was source of strength”: reverse-scored item), and physical neglect (e.g., “Parents were drunk or high”). A high total score indicates respondents had maltreatment experiences. In the present study, the internal consistency for the scale was good (Cronbach’s α = 0.88).

#### 2.2.3. Mediators

Social support was measured using the Social Support Rating Scale (SSRS) [38]. The SSRS is a 10-item scale, measuring subjective support (e.g., “how many friends do you have who are close enough to get support and help”), objective support (e.g., “Please choose sources of financial support and help with practical problems you have received in the past when you have been in a difficult situation”), and the utilization of support (e.g., “Please choose the way you confide your troubles”). A higher total score indicates more social support of respondents. In the present study, the internal consistency for the scale was acceptable (Cronbach’s α = 0.73).

Family functioning was measured using the General Functioning Scale of the Chinese version of the McMaster Family Assessment Device (GF-FAD-C) [39]. The GF-FAD-C is a 12-item, 4-point Likert scale (1 = *strongly agree*, 4 = *strongly disagree*), adapted from the General-Functioning Subscale of the Family Assessment Device, assessing the overall health/pathology of the family (e.g., “there are lots of bad feelings in the family”) [40]. The higher the total score, the worse the overall health of the family is. For ease of understanding, we utilized the revised scores of GF-FAD-C to represent the family functioning. In the present study, the internal consistency for the scale was good (Cronbach’s α = 0.84).

### 2.3. Data Analysis

Data analyses in the present study were conducted using IBM SPSS Statistics for Windows, Version 27.0, IBM SPSS AMOS 24.0, and PROCESS Macro for SPSS 3.5. First, descriptive and correlation analyses were conducted for all variables. Then, common method bias was examined by Harman’s single-factor test. Furthermore, serial mediating roles of family functioning and social support in the relationship between childhood adversity and general distress were examined when controlling for age and gender. To test statistical significance, 95% confidence intervals of bias-corrected boot-strapped method based on 10,000 samples were used.

## 3. Results

### 3.1. Descriptive Analyses and Correlation Analyses

Participant attributes are shown in Table 1. Descriptions and correlations of all variables in the study are shown in Table 2. A total of 2139 participants (996 female and 1143 male) were included in the analysis. The average age of participants was 14.67 years with a standard deviation of 1.53. Childhood trauma was positively correlated with general distress (r = 0.444, *p* < 0.01). Social support was negatively correlated with general distress (r = −0.353, *p* < 0.01). Family functioning was negatively correlated with general distress as well (r = −0.419, *p* < 0.01). Furthermore, childhood trauma was negatively correlated with social support (r = −0.403, *p* < 0.01) and negatively correlated with family functioning (r = −0.566, *p* < 0.01). Social support was positively correlated with family functioning (r = 0.427, *p* < 0.01).

### 3.2. Test of Mediation

Before conducting the mediation analysis, a common method bias test was conducted. The variance of the first factor was 24.27%, less than the critical value (i.e., 40%) examined by Harman’s single-factor test, which indicated no serious common method bias in the present study. The serial mediating model was examined by using SPSS AMOS 24.0 while controlling for gender and age. Table 3 and Figure 1 show the standardized coefficients for total and direct effects on social support, family functioning, and general distress in the serial mediation model. Specifically, childhood trauma was directly associated with general distress with a standardized path coefficient of 0.271. This childhood trauma also had a direct and negative association with social support (standardized path coefficient = −0.406, *p* < 0.001), and a direct and negative association with family functioning (standardized path coefficient = −0.469, *p* < 0.001). The social support had a direct and positive association with family functioning (standardized path coefficient = 0.236, *p* < 0.001), and a direct and negative association with general distress (standardized path coefficient = −0.153, *p* < 0.001). The family functioning was directly associated with general distress (standardized path coefficient = −0.199, *p* < 0.001).

Table 4 shows total, individual, and serial indirect effects for childhood trauma on general distress. The bias-corrected 95% confidence intervals (CIs) precluded zero for indirect effects. Specifically, a significant indirect effect of childhood trauma on general distress via social support was found (indirect effect = 0.062, 95% CI = 0.041–0.184). The effect of childhood trauma on general distress was also mediated by family functioning (indirect effect = 0.094, 95% CI = 0.071–0.117). Furthermore, a significant indirect effect on general distress via social support and family functioning was found for childhood trauma (indirect effect = 0.019, 95% CI = 0.014–0.025). Furthermore, Model 6 of PROCESS Macro for SPSS 3.5 was utilized to conduct the sensitivity analysis and the same results were obtained [41].

Overall, the total effect of childhood trauma on general distress was 0.445, of which 60.7% (0.270) was direct and 39.3% (0.175) was indirect.

## 4. Discussion

The present study examined the association between childhood trauma and general distress among Chinese adolescents, further exploring the underlying mediating roles of social support and family functioning in the linkage of childhood trauma and general distress. Childhood trauma is a traumatic experience which may have a lasting influence on the mental health of adolescents. It is necessary to examine the effect of childhood trauma on mental health first and explore resilience factors for individuals who had childhood adversity in order to minimize the negative consequences.

There are several strengths in the present study. First, the present study takes a perspective of the ecological system to explore the effect of childhood trauma on mental health. Specifically, the mental health of adolescents could be affected not only by adolescents themselves but also by their microsystem (e.g., their family) and mesosystem (e.g., extended family, neighbors, community). Based on this perspective, childhood trauma, family functioning (microsystem), and social support (mesosystem) were considered in the present study to explore the mental health of adolescents. Second, the results of the present study provide further evidence that childhood trauma has an association with mental health in adolescence in a Chinese population. To the best of our knowledge, a limited number of previous studies have been conducted among Chinese adolescents for investigating the relationship between childhood trauma and mental health, although the effect of childhood trauma may be affected by culture and age [6,42]. Third, the results of mediation analysis indicate that the negative consequence of childhood trauma may be long-lasting but may be reduced when there are good social support and family functioning. These resilience factors may shed light on how researchers and practitioners can help adolescents who have had traumatic experiences. Lastly, the present study collected a relatively large sample (*n* = 2139), which may potentially increase the reliability of the results.

Some valuable findings have been obtained in the present study. A positive association between childhood trauma and general distress, with the relationship being mediated both independently and serially by social support and family functioning, was found in the present study.

Specifically, first, in line with previous studies, our results revealed that childhood trauma could positively predict the general distress of adolescents. These previous studies investigated this relationship among adults, college students, juvenile offenders, and community-based adolescents [4,5,6,43,44]. It is preferable to avoid childhood trauma. However, if individuals had been exposed to traumatic experiences during their childhood, investigating the effect of the trauma on mental health in adolescence may be necessary because adolescence is a period of relatively rapid synaptic regrowth compared with adulthood and old age.

Second, we found that social support and family functioning independently mediated the relationship between childhood trauma and general distress, as we hypothesized. Consistent with previous studies, the results showed that childhood trauma could negatively predict social support and family functioning [45]. Moreover, in line with previous studies, we also found that good family functioning and good social support could predict better emotional states [27,46]. Besides, the results supported our hypotheses that childhood trauma would be associated with general distress via social support and family functioning, respectively, which means that childhood trauma has an influence on general distress partially through family and community supports.

Last but not least, as hypothesized, our results showed the serial mediating roles of social support and family functioning in the linkage of childhood trauma and general distress. These results are supported by previous studies. Specifically, both good family functioning and good social support have been confirmed to be resilience factors for individuals who experience childhood trauma [45]. Moreover, social support may have an influence on family functioning due to the following reasons. Individuals who were exposed to a traumatic environment may attempt to seek other resources and supports to disconnect from trauma. These resources and supports may help individuals and their families to cope with the traumatic environment, and further contribute to the family adaption. A previous study found the effect of social support on family functioning in Chinese families of children with Autism Spectrum Disorder, which may provide support for our results [33].

Our findings may have several implications. First, a theoretical implication is that our serial mediation model may be strong evidence and an addition to the ecological system theory of human development and the multisystem developmental framework for resilience. That is, our results revealed that the mental health of adolescents could be affected both by their mesosystem and microsystem. Resilience factors were multisystem, and there are relationships between these resilience factors. Second, our findings may provide evidence-based decision-making for social workers, psychotherapists, and psychological counselors to intervene in the general distress of adolescents who experienced childhood trauma. That is, it is crucial to provide social support to these adolescents and help them increase their family functioning in practical applications. The mental and emotional problems of adolescents who experienced childhood trauma are not merely issues of adolescents themselves, but concerns of the whole system and environment.

Several limitations should be noted in the present study despite the above strengths and implications. First, in the present study the data are cross-sectional, carrying the limitations of the cross-sectional design. That is, we cannot ascertain the causal sequence of variables. It is necessary to conduct longitudinal studies in the future in order to examine the causality of all variables in the present study. Second, subjective self-report and recall bias should be interpreted cautiously because the self-report questionnaire method was utilized in the present study. Questionnaires reported by people who are familiar with the adolescent (e.g., caregivers, best friend) and experiments could be undertaken in the furture in order to examine our results. Third, the present study emphasized the importance of environment on the mental health of adolescents who had childhood trauma, while the effect of individual factors was not investigated. Individual factors (e.g., personality, emotion regulation strategy, cognitive pattern) could be investigated in future studies.

## 5. Conclusions

Returning to the hypotheses posed at the beginning of the present study, it is now possible to state that childhood trauma could positively predict the general distress among Chinese adolescents. Moreover, the social support and family functioning independently and serially mediated the association between childhood trauma and general distress. These findings may contribute to researchers and practitioners getting a better understanding of the emotional problems of adolescents who experienced childhood trauma from an integrated view. Furthermore, our findings indicate that special attention should be given to improvements in social support and family functioning when developing interventions aiming at improving the mental health of adolescents who experienced childhood trauma.

## Figures and Tables

**Figure 1 ijerph-18-06808-f001:**
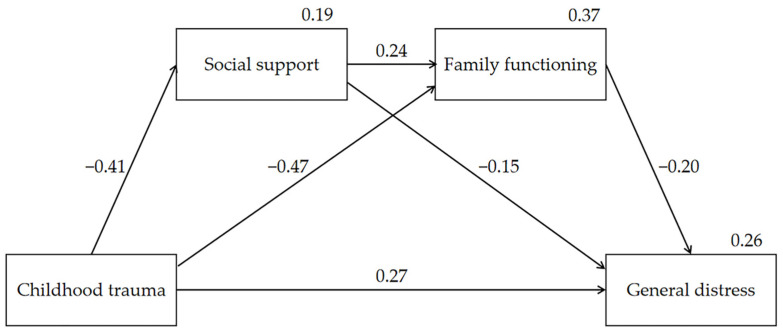
The serial-mediation model showing childhood trauma, social support, and family functioning on general distress.

**Table 1 ijerph-18-06808-t001:** Participant attributes in the present study.

Participant Attributes	*n* (%)/Mean (SD)
Gender	
Male	996 (46.6%)
Female	1143 (53.4%)
Age	14.67 (1.53)
Grade	
7th	698 (32.6%)
8th	712 (33.3%)
10th	300 (14%)
11th	429 (20.1%)

**Table 2 ijerph-18-06808-t002:** Descriptive and correlation analysis.

	Mean	SD	Gender	Age	Childhood Trauma	Social Support	Family Functioning
Gender	--	--	--				
Age	14.67	1.53	0.030	--			
Childhood trauma	34.30	10.36	−0.032	0.016	--		
Social support	38.61	8.59	−0.110 **	−0.113 **	−0.403 **	--	
Family functioning	23.79	6.07	0.018	−0.078 **	−0.566 **	0.427 **	--
General distress	13.80	13.42	0.039	0.056 **	0.444 **	−0.353 **	−0.419 **

*Note. n* = 2139. ** *p* < 0.01.

**Table 3 ijerph-18-06808-t003:** Standardized coefficients for total and direct effects on social support, family functioning, and general distress in the serial mediation model.

Variable	Social Support	Family Functioning		General Distress	
	Total/Direct Effect	Total Effect	Direct Effect	Total Effect	Direct Effect
Childhood trauma	−0.406 ***	−0.565 ***	−0.469 ***	0.445 ***	0.270 ***
Social support			0.236 ***	−0.200 ***	−0.153 ***
Family functioning					−0.199 ***
R^2^	0.188	0.370		0.260	

*Note. n* = 2139. *** *p* < 0.001.

**Table 4 ijerph-18-06808-t004:** Total, individual, and serial indirect effects for childhood trauma on general distress and bias-corrected 95% confidence intervals.

Pathway	Indirect Effect	SE	Bias-Corrected 95%CI	
			Lower	Upper
Total indirect	0.175	0.015	0.147	0.204
Childhood trauma→Social support→General distress	0.062	0.011	0.041	0.084
Childhood trauma→Family functioning→General distress	0.094	0.012	0.071	0.117
Childhood trauma→Social support→Family functioning→General distress	0.019	0.003	0.014	0.025

Note*:* CI = confidence interval.

## Data Availability

The data presented in this study are available on request to the authors. Some variables are restricted to preserve the anonymity of study participants.

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
