# Peer review of "Childhood Trauma and Psychological Distress: A Serial Mediation Model among Chinese Adolescents"

_ijerph, 2021, doi:10.3390/ijerph18136808_

Round 1

Reviewer 1 Report

Overall comments

This is an interesting manuscript reporting on a survey-based study of psychological distress, childhood trauma, social support and family functioning in a large sample of Chinese adolescents. The introduction and discussion could be streamlined a bit (I am not sure the use of subsections is necessary in the introduction and it leads to a bit of repetition). Moderate English editing is needed as there are some grammar and syntax issues.

The study itself is sound though; validated scales are used to measure the constructs of interest, and analyses of the relationships between them are reported comprehensively. Some specific points on each section are included, below:

Introduction

I recommend use of person-centred, trauma-sensitive language, i.e. rather than ‘adolescent victims’ (line 49-50), perhaps something more like ‘people who experienced trauma in adolescence’

Line 101 – it would be helpful to explain what you mean by ‘psychological and physiological ability’ as these are quite broad terms. Also you could explain why you expect (lines 102-106) your findings to accord with those of the studies that have found links between childhood trauma and distress, rather than with the few you reference that predict the opposite. Highlight how your study will help clarify things, specifically.

 The introduction brings in many theories, but is quite descriptive; there is a tendency to very briefly state what each theory predicts/presents, without drilling down into the finer points of the theory, e.g. ‘According to the suboptimal environmental hazards model, childhood trauma can negatively impact the child-parent relationship and family functioning’ (lines 125-127). I think a little more explanation of the theories is needed, if they are to add real value to the rationale being developed (although I do appreciate the challenge in doing this without making the introduction overly long). You could consider dropping/reducing the references to neurobiological theories and findings (in both the introduction and discussion), as although these do have relevance, they are less directly linked to your study which does not delve into the biological side of things empirically.

Methods                               

Please note what timeframe participants are asked to consider in responding to the DASS-21-C (past 12 months or other?).

Similarly, for those unfamiliar with the CTQ-SF-C, specifying the timeframe (is it ‘ever’?) would be helpful.

Results

It is difficult to follow which p-values refer to which correlation with the current use of punctuation in the descriptive analyses (specifically; ‘. Social support (r 231 = -0.353, p < .01) was negatively correlated with general distress and family functioning (r 232 = -0.419, p < .01) was negatively correlated with general distress as well.’ Recommend rewording/changing punctuation for clarity.

The tables and diagrams are helpful and informative.

Discussion

Care should be taken to avoid inferring that the path analysis of cross-sectional data can tell us definitively about causality. For instance, ‘Third, the results of mediation analysis indicate that the negative consequence of childhood trauma may be long-lasting but can be reduced by good social support and family functioning’. I would recommend wording more like ‘are reduced when’ or ‘may be reduced when’, because it is not possible to know from the current study whether manipulating the degree of social support/family functioning will automatically lead to fewer/milder consequences. It is good to see you do address this in lines 341 onwards.

Lines 332-333 you refer to exosystem, but in your results you mention only microsystem and mesosystem. Should this perhaps be ‘mesosystem and microsystem’?

Lines 347-348, it could be perceived as a little dismissive to suggest potential unreliable memory of the adolescent as a limitation – could one also argue that the important thing, in any case, is what adolescent perceives or remembers of their childhood (and possible trauma)?

Line 349, ‘Questionnaires reported by others’ is a little vague and could be expanded on to explain what is meant.

Author Response

We are grateful for taking your time to review this manuscript. We really appreciate all your endorsement, suggestions, and comments. The point-by-point response to the reviewer's comments can be seen in the attachment.

Reviewer 2 Report

The article “Childhood trauma and psychological distress: A serial mediation model among Chinese adolescents” presents an adequate scientific level addressing a topic of interest such as childhood trauma during adolescence. In general, the article is clearly written and easy to follow.

In the following, I propose some improvement adjustment for the work:

  • Add in the introduction the definition of childhood trauma. As for example the one that appears in the following article: Dye, H. (2018). The impact and long-term effects of childhood trauma. Journal of Human Behavior in the Social Environment, 28(3), 381–392.doi:10.1080/10911359.2018.1435328 
  • What criteria were applied to exclude 7% of the questionnaires?
  • Table 1. Add information about the numbers that appear.
  • Data missing in table 2 about the Total effect of Social support.
  • There are no data separated by gender. It might be interesting to observe whether the results vary according to gender. Numerous studies make this differentiation, e.g.:
    • Xiao, D., Wang, T., Huang, Y., Wang, W., Zhao, M., Zhang, W. H., ... & Lu, C. (2020). Gender differences in the associations between types of childhood maltreatment and sleep disturbance among Chinese adolescents. Journal of affective disorders265, 595-602.
    • Pruessner, M., King, S., Vracotas, N., Abadi, S., Iyer, S., Malla, A. K., ... & Joober, R. (2019). Gender differences in childhood trauma in first episode psychosis: Association with symptom severity over two years. Schizophrenia research205, 30-37.
    • Assari, S., & Lankarani, M. M. (2017). Discrimination and psychological distress: gender differences among Arab Americans. Frontiers in psychiatry8, 23.
  • The justification made with the Bronfenbrenner model is limited. For this to be applied, it would be necessary to consider, for example, different cultures or countries where the study is carried out. Otherwise, one can simply deal with two systems (micro and meso).
  • I suggest the authors read the following article to update some of the bibliography they present.: Liu, Q., Jiang, M., Li, S., & Yang, Y. (2021). Social support, resilience, and self-esteem protect against common mental health problems in early adolescence: A nonrecursive analysis from a two-year longitudinal study. Medicine100(4).

Author Response

(The authors gave the same response as above.)

Reviewer 3 Report

Reviewer Report

It is commendable that the author has thoroughly reviewed the previous studies on childhood trauma and clarified the purpose of this study.

As the author states in the latter part of the conclusion, the results of this study can serve as a basis for professional counseling staff to intervene in the general distress of traumatized adolescents.

Minor Comments

1) The author hypothesized that social support might mediate the association between childhood trauma and general distress. The author assessed social support using the Social Support Rating Scale (SSRS). In this study, social support has a significant meaning. The survey and evaluation methods are described, but the specific cases of social support itself are not described at all. So, particular examples of social support related to the subject of this study and the social support envisioned by the author should be indicated.

2) The same applies to family functioning as in the section on social support, and the author should provide specific examples.

3) The authors should be clear about the attributes of the participants. If possible, it should be summarized in a table for easy understanding. In addition, the author should indicate the recruitment method.

4) The author indicates that the number of valid responses was 2139 (93%) of 2300 questionnaires collected from the participants. I think it is essential to show the collection rate, valid response rate, etc.

Author Response

(The authors gave the same response as above.)
